# Nutrient-Response Pathways in Healthspan and Lifespan Regulation

**DOI:** 10.3390/cells11091568

**Published:** 2022-05-06

**Authors:** Aleksandra Dabrowska, Juhi Kumar, Charalampos Rallis

**Affiliations:** School of Life Sciences, University of Essex, Wivenhoe Park, Colchester CO4 3SQ, UK; a.dabrowska@essex.ac.uk (A.D.); j.kumar@essex.ac.uk (J.K.)

**Keywords:** *Drosophila melanogaster*, *Caenorhabditis elegans*, mouse, yeast, translation, transcriptome, proteome, metabolome, TORC1, IGF-1, GWAS, EWAS

## Abstract

Cellular, small invertebrate and vertebrate models are a driving force in biogerontology studies. Using various models, such as yeasts, appropriate tissue culture cells, Drosophila, the nematode *Caenorhabditis elegans* and the mouse, has tremendously increased our knowledge around the relationship between diet, nutrient-response signaling pathways and lifespan regulation. In recent years, combinatorial drug treatments combined with mutagenesis, high-throughput screens, as well as multi-omics approaches, have provided unprecedented insights in cellular metabolism, development, differentiation, and aging. Scientists are, therefore, moving towards characterizing the fine architecture and cross-talks of growth and stress pathways towards identifying possible interventions that could lead to healthy aging and the amelioration of age-related diseases in humans. In this short review, we briefly examine recently uncovered knowledge around nutrient-response pathways, such as the Insulin Growth Factor (IGF) and the mechanistic Target of Rapamycin signaling pathways, as well as specific GWAS and some EWAS studies on lifespan and age-related disease that have enhanced our current understanding within the aging and biogerontology fields. We discuss what is learned from the rich and diverse generated data, as well as challenges and next frontiers in these scientific disciplines.

## 1. Introduction

Aging is the gradual accumulation of changes over time that increases the probability of death; aging is a multifactorial and complex process and an almost universal phenomenon in life [1]. Importantly, the rate of aging and lifespan regulation are amenable and depend on genetic and non-genetic or environmental factors [2]. A significant amount of data has now established that the environment has a profound effect on lifespan regulation, with diet and stress being predominant factors determining survival, at the cellular, tissue and organismal levels [3,4,5,6]. Cells perceive nutrients, i.e., amino acids and sugars, through nutrient-responsive pathways that are hard-wired to basic metabolic processes, such as gene transcription, protein translation, proteostasis and protein degradation rates, mitochondrial function, such as detoxification and respiration, as well as autophagy [2,7]. In this review, we provide a framework of knowledge about the role of nutrient-responsive pathways in lifespan and healthspan (the period of life free of morbidities or pathological conditions) regulation, such as the Insulin Growth Factor and mechanistic Target of Rapamycin (mTOR), and a recent years’ update on the advancements in this scientific field. We briefly refer to fundamental principles of these pathways. In summary, activation of the related signaling controlled by IGF and mTOR, while beneficial early in life, supporting growth and development, seems detrimental in lifespan and healthspan. While the fundamental molecular players around these pathways—although not fully characterized—are sketched on a satisfactory level within simpler organisms, such as yeasts, *D. melanogaster* and *C. elegans*, additional studies are needed to understand functional links on a genome-wide scale. Moreover, single or combinatorial drug treatments that target specific nutrient-responsive and other signaling pathways that affect growth, have been utilized to test effects on lifespan, as well as in healthspan, such as, amelioration of pathological states that might phenocopy age-related diseases or syndromes [8]. 

Nevertheless, genetics, as well as epigenetics, of human aging and the role of diet on human lifespan regulation are still being worked out [9]. The field is utilizing stem cell technologies, patient samples and organoids to bridge this gap and has found itself mature enough to proceed to large studies and clinical trials using mammalian species close to humans, such as dogs [10,11]. However, cross-species comparisons reveal differential tempos, not only in differentiation programs but also within fundamental processes, such as proteostasis and protein half-life patterns, that can affect aging processes and lifespan [12] and healthspan. These studies show that the precise, quantitative outcomes in model organisms might differ from conditions in the human body or even in human cohorts. Therefore, although the contribution of model organisms in biogerontology studies is prolific in understanding underlying molecular mechanisms, interdisciplinary studies combining genetics, biomarker analyses, diet and drug surveys and interventions in human populations are now needed within the field. 

## 2. The Growth Hormone-Insulin Growth Factor Signaling in Lifespan and Healthspan

The role of the somatotropic axis has long been investigated in the context of healthspan and lifespan extension, dating back to the report on the positive impact that human growth hormone (HGH) has on the lean body mass when administered in late adulthood [13] that addresses the former, and a report of dwarf, growth hormone (GH)-deficient mice, addressing the latter [14]. The GH/insulin-like growth factor 1 (IGF-1) axis has been subject to extensive scrutiny, with numerous cellular and animal models being used to establish a positive link between disruption of its components and increased longevity, the details of which have been covered by comprehensive, in-depth reviews [15,16]. Even though the GH-IGF-1 axis is very often broadly described as the key pathway involved in aging, a distinction must be made between the two components. While GH induces hepatic IGF-1 production and the roles of the two hormones are interconnected, the modes of their action do have significant differences and are not synonymous [17]. In this review, we focus on the role of IGF-1 in senescence. Briefly, IGF-1 receptor activation occurs in response to the binding of insulin-like growth factors. Ligands initiate a conformational change in the receptor, stimulating its tyrosine kinase activity, which leads to phosphorylation of a number of different targets, including insulin receptor substrates 1–4 (IRSs) and Src homology collagen (SHC). Src homology 2 (SH2) domain-containing proteins, such as phosphatidylinositol 3-kinase (PI3K) and growth factor receptor-bound 2 (GRB2), recognize phosphotyrosine residues and initiate downstream signaling cascades. Ras/MAPK and PI3K/AKT pathways are the main executors of IGF activation, which ultimately induce expression of transcription factors, upregulate translation and promote cell survival by inhibiting apoptotic signals.

It must be noted that some of the key discoveries in the area of aging research included the identification of insulin/IGF-1 signaling pathway components, which, when inhibited or mutated, extended lifespan in *C. elegans* [18,19,20] and *D. melanogaster* [21,22] and those discoveries paved the way for further exploration of the involvement of this pathway in aging mammals. IGF-1 regulated processes in aging and interventions that target IGF-1 signaling in various model organisms are summarized in Figure 1.

### 2.1. Inhibiting IGF-1 for Increased Health- and Lifespan

The hyperfunction theory of aging suggests that a developmental program acts early in life to promote growth and fertility and, thus, ensures survival during the reproductive peak, continues during later stages of life when it becomes ‘hyper-functional’, leading to cell senescence, development of geriatric diseases and aging [23]. IGF-1 is vital for healthy development but its pleiotropic activity later in life seems to be detrimental to lifespan and some aspects of healthspan. 

The knockdown of the *daf-2* gene (coding for an insulin receptor family member in *C. elegans*) results in increased fitness and longevity, supporting the hyperfunction theory of aging. While targeting *daf-2*, during early life, reduced the fecundity of worms, its downregulation in adulthood increased the lifespan of the organism [24]. Building up on this finding, it has also been shown that adulthood daf-2 deficiency had a positive impact on late-life reproductive potential, as well as the fitness of the offspring [25]. However, is there a time limit on how late such an intervention could be performed and still exert its beneficial effect on longevity? Remarkably, auxin-inducible degradation of daf-2 in *C. elegans* at older ages of the organism still led to the doubling of its lifespan [26]. *daf-2* RNAi treatment in *C. elegans* shed more mechanistic insights, as it revealed reduced SUMOylation of several proteins, one of the targets being car-1, a germline-specific RNA-binding protein. Generation of a mutated form of car-1 that cannot be SUMOylated increased the lifespan of worms [27]. Another consequence of IGF-1 knockdown is a reduction in *cav-1* (coding for caveolin protein) expression, accompanied by a diminished number of neuronal caveolae [28]. This, in turn, seems to reduce toxic protein accumulation and results in extended lifespan. A notable characteristic reported for long-lived *daf-2* mutants is the upregulation of nonsense-mediated mRNA decay, one of RNA quality-control mechanisms [29]. 

Targeting of the IGF-1 receptor (IGF-1R) in mice using monoclonal antibodies results in lifespan extension in females, even when the treatment is performed at advanced ages. Moreover, an improvement in healthspan was reported, as manifested by reduced inflammation and tumorigenesis [30]. Interestingly, a study utilizing ubiquitous inducible IGF-1R knockout (UBIKOR) mice to provide a long-term IGF-1R deficiency model reported maintained healthspan and no cognitive defects. UBIKOR mice had improved mobilization of fat tissue and better adaptation to fasting [31]. 

While the role of IGF-1 in lifespan extension has been extensively studied in different animal models, could it also be applied to predicting life expectancy in humans? Milman et al. looked at IGF-1 serum levels in 184 nonagenarians and found that low-growth-factor levels predicted longer life expectancy in females, as well as in individuals with past incidence of cancer [32]. Moreover, a look into centenarians’ offspring also revealed lower levels of the growth hormone [33]. In terms of healthspan extension, mAbs targeting different components of the IGF signaling pathway have been developed, but currently, the most promising therapeutic targeting IGF-1 and IGF-2 is Xentuzumab. While it did not improve progression-free survival in locally advanced and metastatic breast cancer, a subgroup of patients with non-visceral metastases was identified as potentially benefitting from the treatment, leading to the initiation of a new clinical trial, which is currently ongoing [34]. Moreover, it showed some anti-tumor activity in advanced solid tumor patients [35]. 

While it is feasible to manipulate the genetic backgrounds of animal models or possible to administer antibodies targeting IGF-1R, is there a straightforward way to reduce IGF-1 signaling in a more accessible way? 

### 2.2. Dietary Modulation of IGF-1 Signaling 

In recent years, the popularity of research into ways we can reduce circulating IGF-1 levels has flourished. The discovery that dietary restriction can improve healthspan via downregulation of IGF-1 signaling in different animal models, such as yeast, worms, or rats [36,37,38], has provided promise for a very robust and easy intervention. However, the results of calorie restriction (CR) in primates are not conclusive, with some reporting no change [39], while the studies showing a beneficial effect of CR on lifespan are accompanied by a negative effect on brain matter integrity [40]. While there have been very few long-term clinical studies performed so far, it seems that while no IGF-1 decrease has been reported [41], another insulin/IGF-1 pathway member, IGFBP-1, might play a role, through binding to IGF-1 and a subsequent reduction in its bioavailability [42]. Specific dietary supplementations might also be of importance; for example, selenium supplementation attenuated IGF-1 signaling in mice and improved their healthspan [43]. Phosphatidylethanolamine given to *C. elegans* decreased growth hormone levels, extended lifespan, and reduced oxidative stress. However, long-lived worms displayed reduced fertility, a finding consistent with the hyper-function theory of aging [44]. *D. melanogaster* lifespan extension was achieved with Korean red ginseng, also shown to be mediated by attenuation of IGF-1 signaling, as well as through the histone deacetylase Sir2 [45].

### 2.3. Boosting IGF-1 towards Increasing Healthspan 

The restoration of various IGF-1 signaling proteins has been identified as a mechanism mediating therapeutic effects in various contexts (e.g., platelet-derived biomaterials (PDB) on nicotine-induced intervertebral disc degeneration in mice [46]). Here, we will focus on the effects of IGF-1 in the nervous system.

An organ well known to be affected by age-related IGF-1 decline is the brain. While the role of IGF-1 in various aspects of neural development, plasticity and general pro-cognitive effects have been long established [47], the focus has now shifted to recognizing the impact of IGF-1 decrease on cognitive impairment, as well as other neurological conditions. IGF-1 drop seen in old age became a subject of research and emerging evidence suggests potential therapeutic interventions aimed at boosting cognitive decline and addressing neuronal regeneration via IGF-1 upregulation. Intracerebroventricular (ICV) IGF-1 gene therapy in aging rats improved their spatial memory [48], while inducible, brain-specific overexpression of IGF-1 (bIGF-1) in mice led to an increase in morphological features, such as brain volume and weight, as well as reduced depressive behavior and preserved motor performance [49]. However, while cognitive improvement was not achieved in bIGF-1 mice (characterized by prolonged IGF-1 overexpression), switching to a transient intra-nasal delivery of IGF-1 improved motor learning and memory in aged mice, suggesting that long-term activation can lead to desensitization and loss of cognition-boosting effects. Interestingly, overexpression of a splice variant of IGF-1, mechano-growth factor (MGF), in mice promotes neurogenesis, but only when induced in mature adults rather than aged mice [50]. In a fly model of *C9orf72* repeat expansion, a common culprit of amyotrophic lateral sclerosis and frontotemporal dementia, stimulation of the IGF signaling pathway attenuated the toxic effect of increased poly-GR levels [51].

However, how feasible is it to induce IGF-1 expression in humans? An ideal therapeutic intervention is something that is minimally invasive, cost effective, and easy to implement. Exercise fits this bill perfectly and has been known to increase the uptake of IGF-1 by neuronal sub-groups [52]. Recent research has shown that being more active in old age can lead to improved cognitive function, suggested to be mediated by increased IGF-1 levels [53,54,55]. However, this view is still being debated and not all the research supports the notion that the exercise effects are mediated through IGF-1. While a positive relationship between IGF-1 and hippocampus-dependent memory changes has been reported [56], this effect was independent of exercise, as aerobic fitness regimes did not increase the levels of the growth factor. In another study, it was reported that acute, rather than chronic, exercise led to higher peripheral IGF-1 levels [57]. However, it must be noted that, while in the acute regimen, samples were taken immediately after exercise, in the chronic group, sampling was done within a day. Another study where serum IGF-1 levels were found to be positively correlated with strength or endurance exercise involved taking blood immediately after the intervention [58]. Maass et al. [56] collected samples on the last training day or within one week of the last session. Timing of serum sampling post exercise should be considered when making comparisons, as IGF-1 levels start dropping within the first 24 h of exercise recovery [59]. 

What is the exact mechanism(s) mediating the positive effect of IGF-1 on the brain? The brain is uniquely vulnerable to bioenergetic fluctuations and, thus, to mitochondrial dysfunction [60]. Reduction in circulating IGF-1 in inducible liver IGF-1 knockout mice (*Igf1 f/f*) led to a selective reduction in cortex ATP levels, as well as increased hippocampal oxidative stress, accompanied by an impairment in spatial learning. The drop in the peripheral growth hormone levels did not affect mitochondrial function in muscle fibers, suggesting that it uniquely affects the bioenergetic balance in the brain [61]. Incubation of striatal cells with IGF-1 has also been shown to prevent ROS (Reactive Oxygen Species) formation and improves mitochondrial function in a Huntington’s disease model [62]. It is plausible that the beneficial effects of increased IGF-1 in the environment of the highly energy-dependent brain rely on its role as a positive regulator of mitochondrial biogenesis and dynamics, a function that has been described in other, non-neuronal contexts [63,64,65].

Neurovascular coupling is another key element of healthy brain function. It is a process that links neuronal activity to momentary changes in the cerebral blood flow. *Igf1 f/f* mice have impaired cerebromicrovascular endothelial function, which leads to neurovascular uncoupling and affects cognition. Enhancement of core IGF-1 levels could, thus, have a protective function via the restoration of neurovascular coupling [66]. The structural integrity of cerebral arteries is also affected in *Igf1 f/f* mice with decreasing elastin content and limited adaptation to hypertension [67] and IGF-1 treatment of middle-aged rats prior to cerebral artery occlusion has been shown to improve the integrity of the blood–brain barrier [68].

## 3. The Mechanistic Target of Rapamycin (mTOR) Signaling Pathway in Lifespan and Healthspan Regulation

The mTOR pathway is a well-established, crucial piece of the lifespan puzzle, with numerous studies having successfully shown its implication in lifespan, healthspan and aging [69]. The centerpiece of the pathway is an evolutionarily conserved Ser/Thr-protein kinase, regulating anabolism and catabolism through a plethora of cellular metabolic processes [7] (Figure 1A). There are two mTOR kinases in budding and fission yeasts, while one single kinase is found in mammals. The mTOR kinases function as part of two multiprotein complexes (mTORCs), mTORC1 and mTORC2. The mTORC1 complex is defined by three key subunits: mTOR, Raptor and mLST8. Raptor facilitates substrate recruitment to mTORC1, while mLST8 associates with the catalytic domain of mTORC1, potentially stabilizing kinase loop activation. Additionally, this complex contains two inhibitory units, PRAS40 and DEPTOR [69]. Similarly, the mTORC2 complex also contains mTOR and mLST8, but is differentiated by the presence of Rictor, instead of Raptor [69]. When activated, mTORC1 promotes protein synthesis, mainly through phosphorylation of S6 kinase and the translation regulator 4E-BP1. This complex also induces lipid biogenesis by activating SREBP1 and PPARγ transcription factors. Furthermore, it inhibits catabolism by blocking autophagy [70]. The roles of these complexes in lifespan have been reviewed elsewhere [2,69]. The subcellular distribution and dynamics of mTORC1 complex and components are of great interest. Lysosomes are a crucial point for mTOR-related cellular growth. When inactive, mTORC1 is found within the cytoplasm, while after activation (through nutrients/growth factors), it is recruited to the surface of the lysosome [71]. When bound to GTP, Rheb (Ras homolog enriched in brain, is bound to the outer lysosomal surface) induces a conformational change in mTORC1 resulting in kinase activation in a cellular nutrient-dependent manner [72]. Activity levels of mTOR need to be carefully balanced constantly, as hyperactive functioning of the pathway is linked to multiple diseases, including diabetes, neurodegeneration, and cancer, whereas inhibiting mTOR expression results in an extension of lifespan, as seen in multiple animal models, including mice [69,73]. 

### 3.1. Amino Acid Sensing and mTOR Activity

Dietary amino acids activate mTORC1, with arginine and leucine being well studied and with a lack of both strongly inhibiting mTORC1, as observed in multiple cell types and organisms [74,75]. However, multiple amino acids are implicated in mTORC1 activity regulation; experimental evidence revealed a two-step activation process, wherein certain amino acids, including Asn, Gln, Thr, Arg, Gly, Pro, Ser, Ala and Glu, ‘prime’ or sensitize cells for mTORC1 stimulation, after which ‘activating’-type amino acids, such as Leu, Met, Ile and Val, interact with and promote mTORC1 activation through mTOR phosphorylation. The priming AAs themselves cause little or no direct activation of mTORC1 but are necessary for activating AAs to influence mTORC1 activity [76]. Further research has implicated the role of Sestrin-2 as a leucine sensor interacting with GATOR2, and SLC38A9 as an arginine sensor for the mTORC1 pathway [77,78]. mTORC1 senses amino acid sufficiency, especially leucine and arginine, on the lysosome via Ras-related GTPases (Rag) and arginine through SLC38A9. In addition, arginine, independently of Rag family members, inhibits lysosomal localization of TSC2 to stimulate mTORC1 activity [79]. SAR1B is also reported in the literature as a leucine sensor, involved in mTORC1 signaling. There are some interesting/considerable differences between SAR1B and Sestrin 2: although they both interact with GATOR2, they differ in their binding sites to GATOR2 and their binding affinities to Leucine (SAR1B has a higher binding affinity to leucine, compared to Sestrin-2). Furthermore, in mouse tissues, SAR1B protein was highly expressed in skeletal muscle, whereas Sestrin-2 was abundant in the adipose tissue [80]. Understanding how mTORC1 senses the abundance of these amino acids and carries out its functions is critical in the development of targeted small molecules that can hinder mTORC1-related effects, in a more potent manner. However, these data also highlight multiple pathways that AAs use to interact with mTORC1. Further understanding is still required to design multi-targeted, more successful interventions that could potentially mimic the effects of caloric restriction and inhibit mTOR. 

### 3.2. Rapamycin: Lifespan and Healthspan Regulation

Rapamycin-mediated mTOR inhibition extends lifespan in organismal models, such as *D. melanogaster* and mouse [2,8,81,82,83]. Rapamycin is an anti-fungal agent and an immune suppressant, which also showed lifespan-extending effects in mice by reducing global translation and inducing autophagy [2]. Initial experiments showed a maximal lifespan extension of 9% and 14% in male and female mice, respectively [84]. Rapamycin treatments extend lifespan in mice when treated early in age, as well as in later life, providing evidence that pharmacological interventions could help achieve healthier aging [85]. Effects on lifespan are dose dependent, with greater lifespan extension noted in mice cohorts provided with a higher dose [86]. These data suggest that Rapamycin treatments in humans may be beneficial for lifespan extension. Although no large-scale trials currently exist, a small, randomized, controlled trial has been conducted so far to study feasibility (trials summarized in Figure 2B and presented in Table 1). In an older human cohort of healthy participants (25 participants between 70–95 years old), subjects were provided 1mg of Rapamycin (or placebo) for 8 weeks. The effects of Rapamycin administration included a decline in erythrocytes and blood hemoglobin levels, a slight decrease in the treatment group compared to controls and lower-than-normal levels of certain cytokines, including TNF-α, suggesting that Rapamycin administration could be considered as a safe option in aged humans [87]. Whether or not Rapamycin treatment in humans extends and promotes healthier aging remains to be tested. 

### 3.3. Models for mTOR Studies and Investigation in Humans 

Genetic mouse models containing two hypomorphic alleles for mTOR (*mTOR Δ/Δ*) are used in studies to examine the effects of mTOR expression on lifespan and aging. *mTOR Δ/Δ* mice are viable and tissue analyses reveal 25% expression of wild-type levels of mTOR. These genetic alterations are interesting to study, as they show in vivo effects when both mTORC1 and 2 are negatively affected, as opposed to Rapamycin that primarily inhibits mTORC1. In comparison to controls, this genetic model displayed a downregulation of p16 mRNA expression in tissues, reduced cellular stress and a significant increase in lifespan, in comparison to results from pharmacological interventions [88]. Interestingly, small doses of a pan-mTOR inhibitor, AZD8055, have been tested in vitro on near-senescent skin fibroblasts (HF043). Short-term inhibition of mTORC1 and mTORC2 appear to reverse morphological and biochemical phenotypes of senescence, including reduction in cell size, decreased senescence-related mitochondrial signal, redistribution of lysosomes and a marked decrease in SA-β-gal expression. Clinical trials currently active involving AZD8055 use are limited to cancer; however, results from this study clearly show that it is a promising therapeutic candidate for anti-aging/longevity studies [89]. Outcomes from such studies are helpful to understand the effect of the mTOR pathway on multiple processes that change with aging. However, it is also worth noting that the majority of the studies are conducted using mouse models and translating this into human studies is challenging. 

Larger-scale studies have been conducted using human samples to study mRNA/gene expression of mTOR-related genes. In a cohort of 695 healthy participants from two study groups, mRNA extracted from blood samples was used to study age-associated changes in transcript expression for component genes of the mTOR pathway in human populations. Transcripts relating to the inhibition of translation and initiation (*E1F4EBP2* and *E1F4G3*) were upregulated in both cohorts, whereas most components of S6 kinase were found to be downregulated in one of the two populations. The expression of genes in these cohorts is similar to those seen in laboratory models using mTOR inhibition interventions [90]. Although this is interesting, it would be worthwhile to have similar studies comparing healthy controls and diseased patients, or have cohorts divided by smaller age ranges to explore mTOR transcript ‘signatures’ and what they indicate. Another study looked at gene expression related to the mTOR pathway in nonagenarians, their middle-aged offspring, and age-matched controls. Between 417 nonagenarians and younger controls, *EIF4EBP2* upregulation was once again seen in nonagenarian samples, along with *LAMTOR2*, *AKT1S1*, *PRR5L* and *RHOA*, whereas *FOXO1* and *RAPTOR* expression were lower. The study found *RAPTOR*, *AKT1S1* and *E1F4BP2* to be associated with old age and/or familial longevity, thereby suggesting a transcriptional downregulation of mTORC1 [91]. 

Calorie restriction (CR) studies in humans have shown that it can have a positive effect on aging and health through an increase in metabolism and decrease in oxidative stress [92] (clinical trials using calorie-restriction-related interventions in human aging studies summarized in Figure 2B are shown in Table 2). Since mTOR is a nutrient-sensing pathway, the premise that calorie restriction may cause this positive effect through downregulation of mTOR is theoretically valid. Reducing branched-chain amino acids (BCAA) consumption early on has shown a positive effect on lifespan, specifically in male rats, with a 34.9% median increase and 18.2% overall increase in maximum lifespan. However, this effect was not seen in females. Transcriptional profiling of quadricep muscles from the male cohort showed a significant inhibition of mTORC1, reduction in phosphorylation of mTORC1-related substrates and an upregulation of negative regulators of mTORC1, including Sesn2 and CASTOR1 [93]. Components of the IGF-1 and mTOR pathways are known with different names in the model organisms used, including humans. The basic components are summarized in Figure 3.

## 4. Human Aging, Lifespan, and Healthspan Analysis Studies

Nutrient-responsive pathways, such as mTOR and IGF, are directly implicated in disease and cancer and numerous clinical trials involving related inhibitors (e.g., rapalogues) are currently under way (www.clinicaltrials.org, accessed on 17 March 2022). Towards delineating human aging regulation and identifying genetic, epigenetic, and environmental factors affecting aging and lifespan, significant efforts have been focused on large surveys: biomarkers analysis in aged versus young human cohorts, investigation using samples from centenarians and over-centenarians, Genome-Wide Association Studies (GWAS) and more recently, Epigenome-Wide Associated Studies (EWAS). For these analyses, the cutoffs of the human cohorts used are arbitrary and differ from each other. This is an issue in current Biogerontology, together with a lack of concrete and standard terminology and universal methodology of population stratification for such studies. Admittedly, the approaches are not simple and are realistically dictated from the availability of appropriate individuals. In this section, we briefly discuss related results from such approaches and whether they link to dietary factors and nutrient-responsive pathways.

### 4.1. GWAS Studies on Lifespan and Healthspan

Initial GWAS studies on human longevity included small cohorts, accompanied, however, with appropriate and multiple statistical tests. The studies point towards specific loci related to apolipoprotein E (APOE), an angiotensin-converting enzyme (ACE) that might be related to lifespan [94]. Subsequent GWAS studies on age-related diseases, such as Alzheimer’s (AD), have linked related loci to healthspan [95,96]. APOE variants are strongly highlighted in multiple studies with E4 increasing, whereas E2 decreases the risk of late-onset AD compared with the E3 variant. APOE4 constitutes the most important genetic risk factor for Alzheimer’s disease (AD) and it impairs microglial function and impedes astrocytic Aβ clearance in the brain [97,98,99]. APOE2 protective mechanisms against AD are under scrutiny and include both amyloid-β (Aβ)-dependent and independent processes [100]. Importantly, APOE2 has been identified as a longevity gene, but curiously, E2 carriers exhibit increased risks of certain cerebrovascular diseases and neurological disorders [100,101,102]. 

A significant number of results stems from GWAS studies using large cohorts, derived from the UK Biobank, including a variety of deep genetic and phenotypic data, collected on approximately 500,000 individuals from across the United Kingdom, aged between 40 and 69 at recruitment [103], in combination, or not, with large data from consortia, such as LifeGen [104] and AncestryDNA [105]. Large GWAS analyses have concentrated on two main approaches: firstly, connecting parental lifespan with healthspan and lifespan of the progeny and, secondly, direct comparisons of cohorts comprising long-lived individuals compared with younger individuals.

A series of parental GWAS analyses based on UK Biobank have reported loci related to human lifespan. Firstly, a study including 75,000 individuals reported, again, APOE alleles as associated with longevity. The study also reported a series of protective alleles for age-related diseases (cardiovascular, metabolic diseases) in the progeny of longer-lived parents. This is a characteristic of such studies, with disease-associated alleles more easily revealed compared with direct longevity-controlling variants. Subsequent parental age study, including 389,166 individuals aged 40 to 73 years old [106] beyond APOE, reported associations with the *CHRNA3/5* (nicotinic acetylcholine receptor) locus (rs1317286). Associations were also found for six other loci already implicated in disease-specific GWAS, such as rs55730499 in an intron of the *LPA* gene, associated with lipoprotein A and LDL (Low Density Lipoprotein) cholesterol levels. Variants in the 9p21 region, which include the long non-coding RNA *CDKN2B-AS1* (*ANRIL*) and *CDKN2A/B*, were also associated. This locus is of particular interest in aging and age-related diseases. Multiple SNPs are associated with specific diseases, displaying cell or tissue-specific implication in senescence, type-2 diabetes and multiple cancers. Notably, most of the SNPs are associated with non-coding rather than coding regions (the locus contains protein coding genes for p14, p15 and p16) [9,106,107]. In another study, using genome-wide association meta-analysis of 606,059 parents’ survival, previous suggestions, such as *APOE*, *CHRNA3/5* and *CDKN2A/B,* and two more regions, namely *HLA-DQA1/DRB1* (coding for HLA class II beta chain paralogue) and *LPA* (coding for Lipoprotein A), are linked to longevity [108]. While these associations relate to both maternal and paternal lifespans, specific variants are found to be associated with either maternal or paternal lifespans only. For example, the paternally associated variant *rs15285 (G)* in lipoprotein lipase is linked with cardiometabolic disease [9]. Parental lifespan GWAS studies are enhanced by studies comparing cohorts of old versus young individuals, with this distinction being arbitrary and varied between studies. A recent study that contained two meta-analyses of GWAS included 11,262 and 3484 cases surviving at or beyond the age corresponding to the 90th and 99th survival percentile, respectively, and 25,483 controls whose age at death or at last contact was at or below the age corresponding to the 60th survival percentile [109]. Consistent with previous reports, the APOE2 variant was linked to longevity while APOE4 with lower odds of surviving to the 90th and 99th percentile age [109]. 

Beyond APOE and CDKN2A/B-related variants, one of the most robust loci in lifespan GWAS studies is FOXO3A [110]. There are strong connections of FOXOA3 or its orthologues (such as daf-16 in *C. elegans*) with nutrient-responsive pathways and diet. *Daf-16* gene encodes a member of the hepatocyte nuclear factor 3 (HNF-3)/forkhead family of transcriptional regulators and plays a key role in modulating the effects of the IGF pathway in healthspan and lifespan [111,112]. Since the first studies on *daf-16* emerged, FOXO transcription factors have been intensively studied and have been established as master regulators of lifespan, with close connections to metabolism, genome integrity and nutrient-responsive pathways, such as mTOR and IGF [113]. The association of a *FOXO* gene with human lifespan makes it very plausible that the mechanisms of action of these transcriptional regulators might be preserved and apply to human organismal aging as well. 

Another locus of interest is the one containing the gene *SH2B3* and nearby genes (*ATXN2* and *BRAP*). *SH2B3* encodes for the ubiquitously expressed lymphocyte adaptor protein LNK, a regulator of signaling pathways relating to hematopoiesis [114], inflammation, and cell migration, and implicated in related diseases [108,115]. Variants are related to differential lifespans (shorter or longer), as well as associated autoimmune and cardiovascular conditions [116] and various cancers, including myeloproliferative cancers, as well as breast, colorectal and lung cancers [117]. SH2B in *D. melanogaster* directly binds to Chico (homologue of vertebrate insulin receptor substrate, IRS), promoting insulin-like signaling [118]. Loss of the *SH2B* orthologue in *D. melanogaster* results in longer lifespans during starvation conditions [118,119], while it is implicated in neurodegeneration and accumulation of β-amyloid [120]. Similarly, mouse and human SH2B1 and SH2B2 are related to glucose homeostasis [121], the insulin signaling pathway [122,123]. In mice, glucose tolerance and insulin responses are impaired in *Lnk/Sh2b3*^−/−^ mice and high-fat diet worsened their glucose intolerance [124]. The examples of *FOXO3* and *SH2B3* illustrate the identification of lifespan and healthspan candidate genes that are shown to be implicated with nutrient-response pathways in model animals, supporting the notion that these pathways are major players in these processes in humans.

### 4.2. EWAS Studies in Human Lifespan and Healthspan: Links to Diet?

DNA methylation is the most studied epigenetic mark, and it has been linked to organismal and tissue-specific aging hallmarks and senescence [125,126,127], with methylation patterns changing over time and scientists introducing the idea of an epigenetic biological clock [128]. There are numerous EWAS studies that have linked methylation to specific diseases while others have connected DNA methylome lifestyles [129,130]. Studies looking at differentially methylated regions in twins for aging phenotypes, as well as longevity, concluded that most age-related changes in DNA methylation are not associated with phenotypic measures of healthy aging in later life. Nevertheless, this EWAS study identified hundreds of regions that are differentially methylated during aging [131]. Many EWAS studies are conducted for age-related pathologies or lifespan. However, what about EWAS studies that could provide a direct link of nutrition or diet to aging or age-related diseases in humans? 

The neuroactive methylxanthine compound caffeine, reported as an mTOR inhibitor, has been linked to increased chronological lifespan (CLS, defined as the time that a postmitotic population is viable), as well as protective effects against diseases, such as cancer and improved responses to clinical therapies [132,133]. DNA methylome EWAS analyses on coffee and tea consumption in 15,789 participants of European and African-American ancestries from 15 cohorts revealed 11 significant CpGs, located near the *AHRR*, *F2RL3*, *FLJ43663*, *HDAC4*, *GFI1* and *PHGDH* genes (the latter coding for phosphoglycerate dehydrogenase) [134]. One of the candidate hits was significantly associated with expression of the *PHGDH* and risk of fatty liver disease and further knock-down experiments of *PHGDH* in liver cells pointed towards a role in hepatic-lipid metabolism. These results link a nutrient and possible direct, or indirect [133], mTOR inhibitor, in possible age-related metabolic pathologies. In another diet-related EWAS study, blood-based differential methylomes comparing responders and non-responders to vitamin K1 supplementation (identified in a 3-year supplementation test) revealed multiple regions with previously unknown relationships to Vitamin K1 absorption and metabolism, such as at the *TMEM263* locus [135], coding a gene previously reported to be involved in skeletal dysplasia [136]. Finally, in another study, differential methylome was examined after bariatric surgery. There was a decrease in differentially methylated CpG sites by 51%, while gene enrichment analysis indicated processes including regulation of transcription, RNA metabolic, and biosynthetic processes [137]. 

## 5. Conclusions

We briefly summarized data regarding the IGF-1 and mTOR signaling pathways and presented large GWAS and EWAS efforts that have highlighted loci associated with lifespan regulation, as well as in age-related diseases. Our coverage of pathological states is not extensive, and we have not covered cancer, as this would be beyond the scope of this brief review. The most prominent loci linked to longevity from GWAS studies are related to the APOE2 variant and the long non-coding RNA *ANRIL*. Nutrient-response pathways are key to understanding the basic biology of aging and cellular physiology. Diet impacts our basic metabolism on a cellular and organismal level and affects microbiota, bioenergetics, growth, differentiation, healthspan, and ultimately, lifespan. Differential gene expression studies have shown that nutrient perception genes are key with, for example, *RAPTOR*, *AKT1S1* and *E1F4BP2* being associated with old age and/or familial longevity, suggesting a transcriptional downregulation of mTORC1 [91]. In addition to APOE2 and *ANRIL*, GWAS studies have pointed to the relation of nutrient response with human lifespan; FOXO3A, a transcription factor related to IGF signaling, is a significant locus, robustly appearing in such studies. More GWAS studies are conducted for age-related diseases than lifespan itself, while EWAS studies are even less. Large scale epidemiological types of studies are needed for the field, covering from biochemical assays/biomarkers to gene expression and metabolome changes in a longitudinal way. Such studies should include defined dietary interventions, and could potentially provide quantitative data on human aging rates and the control of age-related diseases. While basic research on the biology of aging is crucial and necessary to fuel the field of biogerontology with ideas and unique insights, interdisciplinary large projects that would span molecular, cellular, organismal and population levels will be a defining next step. 

## Figures and Tables

**Figure 1 cells-11-01568-f001:**
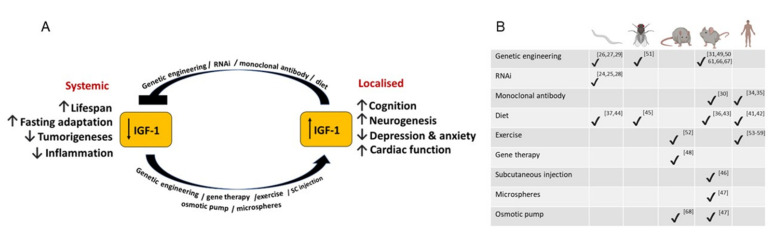
Schematic of IGF1-regulated processes in aging (**A**) and a breakdown of IGF-1-modulating interventions applied to each of the animal models used in the reviewed studies (**B**). Numbers in panel indicate references related to the corresponding interventions in the various model organisms.

**Figure 2 cells-11-01568-f002:**
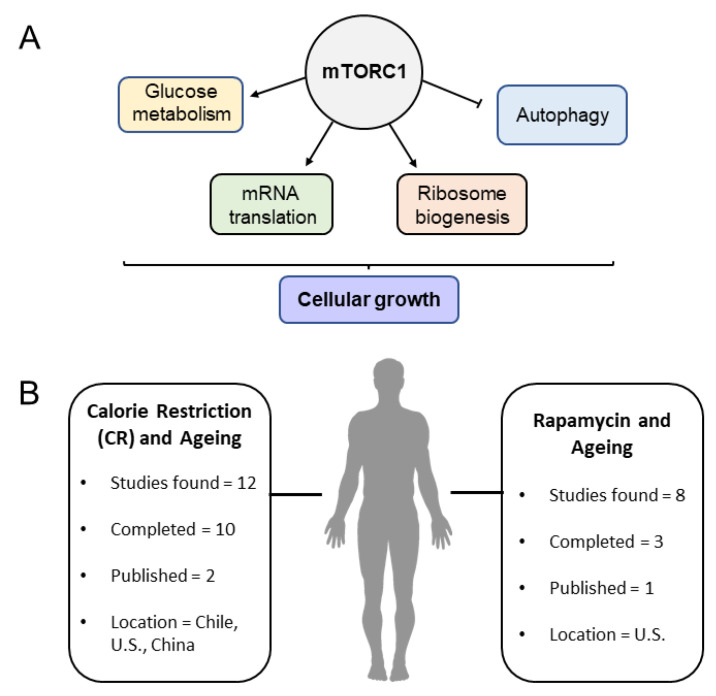
Schematic of mTORC1-regulated processes (**A**) and summary of clinical trials related to mTORC inhibition and calorie restriction (**B**).

**Figure 3 cells-11-01568-f003:**
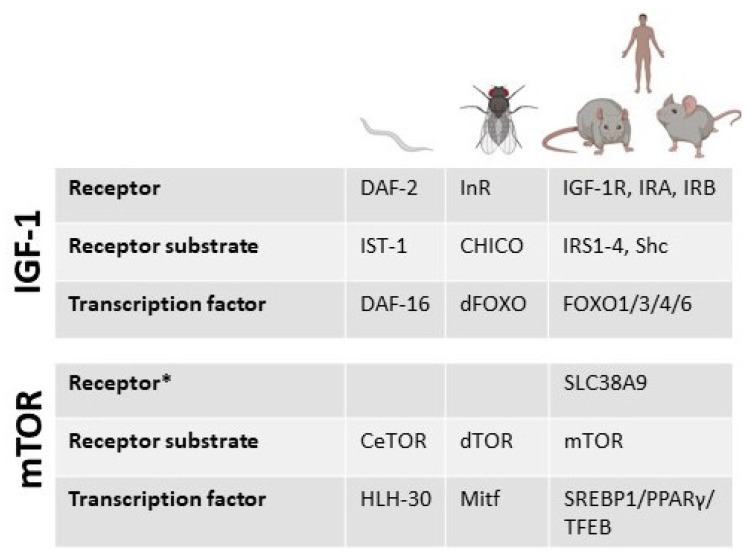
Schematic showing different components of the IGF-1 and mTOR signaling in model organisms. * While cell membrane receptors for IGF-1 are presented for mTOR we include a lysosomal receptor reported in the literature and mentioned within the text.

**Table 1 cells-11-01568-t001:** Summary and information on clinical trials using Rapamycin-related interventions in human aging studies (summarized in Figure 2B) including study status, interventions and number of participants. Information gathered from www.clinicaltrials.gov, accessed on 17 March 2022.

Study Title and Phase	Status	Interventions	Enrollment (Estimated)
Participatory Evaluation (of) Aging (With) Rapamycin (for) Longevity Study [Phase II]	Recruiting	Rapamycin and placebo	150 participants
The Role of Sirolimus in Preventing Functional Decline in Older Adults [Phase II]	Not yet recruiting	Sirolimus	14 participants
VIAging Deceleration Trial Using Metformin, Dasatinib, Rapamycin and Nutritional Supplements [Phase I]	Not yet recruiting	Study drugs (including Rapamycin) and nutritional supplements	50 participants
Effect of mTOR inhibition and Other Metabolism Modulating Interventions on the Elderly [Phase II]	Completed—has results	Rapamycin and placebo	34 participants
Effect of mTOR Inhibition and Other Metabolism Modulating Interventions on the Elderly (Substudy Rapa and cMRI to Evaluate Cardiac Function) [Phase II]	Recruiting	Rapamycin	12 participants
Exercise and Low-Dose Rapamycin in Older Adults With CAD: Cardiac Rehabilitation and Rapamycin in Elderly (CARE) trial [Phase I]	Completed—no results available	Rapamycin	13 participants
Phase I Study of the Effects of Combining Topical FDA-approved Drugs on Age-related Pathways on the Skin of Healthy Volunteers	Completed	Sirolimus, Metformin, Diclofenac	10 participants
Topical-RAPA Use in Inflammation Reversal and Re-setting of the Epigenetic Clock [Early Phase I]	Active, not recruiting	Rapamycin topical ointment and placebo	50 participants

**Table 2 cells-11-01568-t002:** Information on clinical trials using calorie-restriction-related interventions in human aging studies (summarized in Figure 2B) including study status, interventions and number of participants. Information gathered from www.clinicaltrials.gov, accessed on 17 March 2022.

Study Title and Phase	Status	Interventions	Enrollment (Estimated)
Calorie Restriction Retards the Aging Process	Unknown	Energy restricted Mediterranean-type diet; 25% calorie restriction	48 participants
The Effect of Food Stimuli on the Calorie Restriction Response in Healthy Subjects	Completed	Stimuli of food smell and vision vs. no stimuli of food smell and vision	12 participants
Effect of Age and Weight Loss on Inflammation and Iron Homeostasis	Completed	Calorie restriction	44 participants
Effect of Resvida, a Comparison with Calorie Restriction Regimen	Completed	Resveratrol, placebo, and calorie restriction	58 participants
Metformin Induces a Dietary Restriction-like State in Human [Phase IV]	Unknown status	Metformin 0.85 twice daily for six months, calorie restriction	60 participants
CALERIE Phase II Ancillary: Metabolic [Phase II]	Completed	Calorie Restriction	75 participants
CALERIE (Tufts)—Comprehensive Assessment of Long-Term Effects of Reducing Intake of Energy [Phase I]	Completed	Calorie Restriction	44 participants
Long-term Caloric Restriction and Cellular Aging Markers	Completed—Has results	No interventions	71 participants
CALERIE (Washington University): Comprehensive Assessment of Long-Term Effect of Reducing Intake of Energy [Phase I]	Completed	Calorie Restriction	48 participants
The Effect of Time-restricted Feeding on Physiological Function in Middle-aged and Older Adults [Phase I/II]	Unknown status	Time-restricted feeding	12 participants
CALERIE: Comprehensive Assessment of Long-Term Effects of Reducing Intake of Energy	Completed—has results	Calorie restriction and control	238 participants
CALERIE (PBRC, Baton Rouge)—Comprehensive Assessment of Long-Term Effects of Reducing Intake of Energy	Completed	Calorie restriction and exercise	48 participants

## Data Availability

Not applicable.

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
