# Peer review of "Nutrient-Response Pathways in Healthspan and Lifespan Regulation"

_cells, 2022, doi:10.3390/cells11091568_

Round 1
Reviewer 1 Report
Manuscript Cells-1671986 entitled " Nutrient-response pathways in healthspan and lifespan regulation" by Aleksandra Dabrowska et al. is a review article and its should be summarizing the recently discovered related to the regulation of metabolic health and aging by nutrient-sensitive signaling pathways.
The manuscript was hastily drawn up. It is not friendly in reading. Some improvements are needed in this work. Also, a better abstract and conclusion should be provided.
The introduction is poor. Authors must remove the conclusions from the introduction.
The discussion of the various paragraphs is fragmented.
Table 2 cannot be read.
This paper needs extensive improvement to achieve the standard of quality needed in the Cells journal publications
Author Response
Reviewer 1
The manuscript was hastily drawn up. It is not friendly in reading. Some improvements are needed in this work. Also, a better abstract and conclusion should be provided.
We have included edits throughout the manuscript that help improving the flow and making it more accessible. However, the other reviewers are happy with the quality of writing and the content.
The introduction is poor. Authors must remove the conclusions from the introduction.
We have done editing amendments in the introduction although it is not very clear why the reviewer thinks the introduction as poor. We have amended the paragraph of the conclusion.
The discussion of the various paragraphs is fragmented.
This has to do with the diverse content of the review. We have now amended our conclusion in order to provide a more unified message for the manuscript. We hope that this is now satisfactory.
Table 2 cannot be read.
The table follows the format of the journal. We thank the reviewer for the input but the comment is vague and the other reviewers do not have a problem with reading the table.
This paper needs extensive improvement to achieve the standard of quality needed in the Cells journal publications
Please see point 1. We have done edits throughout to improve the manuscript.
Reviewer 2 Report
The review by Dabrowska et al. discusses the aging-related effects of the nutrient-sensing GH/IGF1 and mTOR pathways and the roles of the genes discovered in the GWAS and EWAS studies centenarians. The authors summarize the knowledge of how these metabolic pathways regulate aging and cognition. The manuscript combines knowledge from several model organisms - I find it particularly useful, as most current reviews on this topic typically focus on a single animal model. Nevertheless, this challenging approach would benefit from a clearer explanation of the phylogeny and structure of the discussed pathways. Otherwise, the review is well written and easy to follow.
I have the following suggestions:
1. It would be helpful to briefly explain the IGF1 and TOR pathways on a molecular level. Consider including a schematic figure showing the pathways in the mentioned animal models. The homologous genes sometimes have unrelated names in different animals, such as Daf2 in the worm, InR in the fly, and IGF-1R in mice. Without a good explanation/scheme, a naïve reader may get confused about how the discussed data are related.
2. Figure 1B and Table 1: Please insert reference numbers into the table.
3. A lot is known about the aging-related roles of Insulin/insulin-like growth factor signaling in the fly model, and at least some of the key studies should be discussed (not only the referenced). The same goes for the rapamycin and Tor pathway.
Moreover, Drosophila has been particularly useful in studying the relations between obesity, aging, nutrient-sensing pathways, and diets. Perhaps you should mention this and consider discussing some of the key findings - especially since Drosophila is listed as the first keyword.
4. The abstract promises that the authors ‘examine recently uncovered interplays of nutrient-response and stress pathways with basic cellular metabolism and bioenergetics…’ (lines 16-17). However, while the aging-related effects of the nutrient-sensing pathways are discussed in detail, there is almost nothing on the metabolic roles of the pathway. Therefore, I suggest adjusting the abstract or including a corresponding chapter.
Minor comments:
- Line 9: use a capital C: C. elegans, not c. elegans
- Line 38 and elsewhere: When mentioning several animal models in one sentence, consider using the same form, either the abbreviated genus + species name (e.g., C. elegans and D. melanogaster) or just the genus name (Caenorhabditis and Drosophila). I would not mix the styles (just for consistency).
Author Response
Reviewer 2
- It would be helpful to briefly explain the IGF1 and TOR pathways on a molecular level. Consider including a schematic figure showing the pathways in the mentioned animal models. The homologous genes sometimes have unrelated names in different animals, such as Daf2 in the worm, InR in the fly, and IGF-1R in mice. Without a good explanation/scheme, a naïve reader may get confused about how the discussed data are related.
We have now added this information. The changes within the manuscript are tracked as required. We have added a new figure (Figure 3) to include information that the reviewer has suggested.
- Figure 1B and Table 1: Please insert reference numbers into the table.
We appreciate this comment. We have added the reference numbers in the appropriate positions.
- A lot is known about the aging-related roles of Insulin/insulin-like growth factor signaling in the fly model, and at least some of the key studies should be discussed (not only the referenced). The same goes for the rapamycin and Tor pathway.
We do comment that the pathways have been extensively reviewed elsewhere but we have added some general aspects in order to help the readers. We have also added Figure 3 that can be used for easy access to this information.
Moreover, Drosophila has been particularly useful in studying the relations between obesity, aging, nutrient-sensing pathways, and diets. Perhaps you should mention this and consider discussing some of the key findings - especially since Drosophila is listed as the first keyword.
We have added related references that have revealed these processes in Drosophila and have included relevant players of the pathways in Figure 3.
- The abstract promises that the authors ‘examine recently uncovered interplays of nutrient-response and stress pathways with basic cellular metabolism and bioenergetics…’ (lines 16-17). However, while the aging-related effects of the nutrient-sensing pathways are discussed in detail, there is almost nothing on the metabolic roles of the pathway. Therefore, I suggest adjusting the abstract or including a corresponding chapter.
We have adjusted the abstract accordingly.
Minor comments:
- Line 9: use a capital C: C. elegans, not c. elegans
This is amended as required
- Line 38 and elsewhere: When mentioning several animal models in one sentence, consider using the same form, either the abbreviated genus + species name (e.g., C. elegans and D. melanogaster) or just the genus name (Caenorhabditis and Drosophila). I would not mix the styles (just for consistency).
This is amended accordingly. We use throughout the species names.
Reviewer 3 Report
The article entitled “Nutrient-response pathways in health span and lifespan regulation” is a review on new insights on the connection between nutrient-response and the stress pathways with cellular metabolism underlining the connection with pathological aspect associated to ageing. Even if these aspects are well characterized in model organisms, the authors evidenced that they could not be exhaustive for the human body, underlying, in this case, the necessity of focused approach which combines genetics, biomarker analyses, diet and drug surveys. In this context interesting comparisons are made between the studies on animal models and the analyses obtained in human research.
The article is well written with a good part of bibliography of recent years, so the arguments on the article are sufficiently updated. In my opinion also the topic is well focused and different aspects are also well addressed.
Author Response
Reviewer 3
The reviewer is happy with the manuscript and has raised no concerns or comments.
Round 2
Reviewer 1 Report
A mio parere, la qualità tecnica di questa carta è sufficiente. Consiglio di ridurre le dimensioni della tabella 2 perché l'ultima colonna non è completamente leggibile (impostarla come Tabella 1)
Reviewer 2 Report
I think the authors have considerably improved their manuscript. I do not have further comments or requests.